# Molecular Basis of Resistance to Mesosulfuron-Methyl in a Black-Grass (*Alopecurus myosuroides* Huds.) Population from China

Xiangyang Qin [1], Cheng Yang [1], Mengmeng Hu [1], Yunxia Duan [1], Na Zhang [2], Jinxin Wang [1], Hengzhi Wang [1,*] and Weitang Liu [1,*]

[1] College of Plant Protection, Shandong Agricultural University, Tai'an 271018, China
[2] Research Center of Pesticide Environmental Toxicology, Shandong Agricultural University, Tai'an 271018, China
* Correspondence: wanghz@sdau.edu.cn (H.W.); liuwt@sdau.edu.cn (W.L.)

**Abstract:** Black-grass (*Alopecurus myosuroides* Huds.) is a common weed in Chinese wheat fields, and has become troublesome due to its evolution of herbicide resistance. One black-grass population (HN-14) collected from a wheat field where herbicides were applied was suspected to be resistant (R) to mesosulfuron-methyl. This study aims to establish a cross-resistance pattern and explore potential resistance mechanisms. The results of a whole-plant dose response assay showed that the resistant (R) population had a high of resistance to mesosulfuron-methyl (33-fold); meanwhile, no synergism of P450s activity inhibitor malathion was observed. The sequencing results revealed that ALS resistance mutation Trp-574-Leu occurred in R plants. The results of in vitro ALS enzyme activity assays also supported that the extractable ALS from R plants were 23.22-fold resistant to mesosulfuron-methyl. In the light of the "R" resistance rating system, HN-14 has evolved RRR and RR resistance to fenoxaprop-P-ethyl, clodinafop-propargyl, haloxyfop-methyl, and fluazifop-P-butyl and R? (resistance may be developing) to pinoxaden, however remains sensitive to imazethapyr, quizalofop-P-ethyl, tralkoxydim, and isoproturon. These results indicated that the mesosulfuron-methyl resistance in the black-grass population HN-14 was driven by a target-site mechanism rather than a nontarget (at least P450s-mediated) mechanism.

**Keywords:** herbicide resistance mechanism; cross resistance; acetolactate synthase (ALS) inhibitor; *Alopecurus myosuroides* huds



## 1. Introduction

Black-grass (*Alopecurus myosuroides* Huds.) is a main weed of grains in northwest Europe [1]. At first, it was only found in Taiwan, China, but now its problems are becoming more and more serious in mainland China, including in Shandong, Henan, and Anhui Provinces. The emergence of herbicide resistance has an important impact on wheat production [1,2]. Since 1982, when it was reported that *Lolium rigidum* populations in Australian wheat fields were tolerant to the herbicide diclofop-methyl, the resistance to ACCase inhibiting herbicides has continued to rise [3]. So far, it has been reported that 48 weeds in the worldwide are resistant to ACCase inhibitors, including 7 species in China [4]. After ACCase inhibitors failed to control weeds of grains, mesosulfuron-methyl was applied as a substitute in these areas disturbed by ACCase-resistant grass weeds. Mesosulfuron-methyl, a sulfonylurea (SU) herbicide, can inhibits the activity of the acetolactate synthase (ALS, EC 2.2.1.6) and catalyzes the first step of biosynthesis of branched-chain amino acids (isoleucine, leucine, and valine) [5]. ALS-inhibiting herbicides have low toxicity to mammals, high activity, and a broad spectrum of weed control. ALS-inhibiting herbicides have been widely and repeatedly used since the early 1980s [6,7].

Unfortunately, to this day, since ALS inhibitors are prone to develop resistance, 160 weeds have developed resistance to them [4].

In addition to the use of ACCase- and ALS-inhibiting herbicides, non-chemical weed control in wheat is recommended. Vandersteen et al. [8] reported that the reproduction rate of black-grass can be reduced by 40% after sowing is delayed for two weeks. Planting spring wheat and using template plough can reduced black-grass in wheat fields by 88% and 69%, respectively [9]. Field cultivation, competitive crop varieties, and higher sowing rates are additional solutions to resistance to single herbicides [10]. However, non-chemical measures are not conducive to the spread of weed seeds and thus can reduce the number of weeds in the coming year, they cannot reduce the proportion of resistant individuals in the whole when resistance transmission changes [11]. Therefore, in order to be able to control the sustainability of black-grass, we must better unravel the reasons and processes generated by the herbicide resistance evolution.

Target-site resistance (TSR) and non-target-site resistance (NTSR) are two mechanisms involved in weed resistance [12]. In most cases, TSR is caused by the mutation(s) of target genes, which reduce(s) enzyme sensitivity to herbicides [12]. At least 29 amino acid mutations located in 8 positions in the ALS gene have been reported to repeatedly endow resistance in grass weeds: Ala122, Pro197, Ala205, Asp376, Arg377, Trp574, Ser653, and Gly654 [4,12]. NTSR is another powerful resistance mechanism, and has been frequently reported in herbicide(s) resistance in recent years. Metabolic resistance is one of the main NTSR mechanisms, which often involves cytochrome P450 monooxygenase, glutathione S-transferase, and glycosyl transferase [13]. The sustainability and crop production of metabolic-resistance-threatened herbicides are due to limited knowledge of their mechanisms, and the unpredictable cross-resistant mode. Many studies have globally reported the resistance of black-grass to ACCase and ALS inhibitors [14–19]. In recent years, more and more populations have been identified as herbicide-resistant in China. What the attention should be caused is that the multiple herbicide resistance (MHR) in these *Alopecurus* sp. weeds. In China, Bi et al. [20] reported that an *Alopecurus japonicus* population had different degrees of resistance to fenoxaprop-P-ethyl and mesosulfuron-methy, because Ile1781 in ACCase and Trp 574 in ALS were replaced by Leu, respectively. Guo et al. [21] reported that the resistance of an *Alopecurus aequalis* population to both fenoxaprop-P-ethyl and mesosulfuron-methyl because of Ile-2041 in ACCase and Pro-197-Arg in ALS mutations, respectively. Lan et al. [22] reported that a black-grass AH93 population was cross-resistant to mesosulfuron-methyl and multiple-resistant to pinoxaden. Phe-206-Tyr mutations were discovered in black-grass that was resistant to mesosulfuron-methyl, flucarbazone-sodium, and imazethapyr [23]. However, MHR in black-grass has been less frequently reported. In a previous study, we identified one black-grass population (HN-14) that exhibited resistance to fenoxaprop-P-ethyl and mesosulfuron-methyl. However, the resistance status to mesosulfuron-methyl, the resistance mechanism, and the cross-resistance pattern were still unknown. Thus, the purposes of this study were (1) to establish the resistance levels to mesosulfuron-methyl in this specific population; (2) to investigate the mechanism involved in mesosulfuron-methyl resistance; (3) to identification of cross resistance modes of other herbicides under different action modes.

## 2. Materials and Methods

### 2.1. Seed Collection in the Fields

In this study, two black-grass populations (HN-14 and HN-06) were used. The population HN-06 was known to be sensitive to herbicides, and HN-14 was confirmed to be resistant to fenoxaprop-P-ethyl in our previous report [24]. Detailed information on the two populations is shown in Table 1. All the seeds were air-dried and stored in sealed plastic bags at 4 °C.

**Table 1.** Collection information and resistance of black-grass-seeds.

| Population | State | Location | | | Longitude/Latitude | Site | Time | Numbers of Herbicide Application |
| | | Village | Country | Province | | | | |
| --- | --- | --- | --- | --- | --- | --- | --- | --- |
| HN-14 | R [a] | Luodian | Zhumadian | Henan | N33.056796 E114.121244 | Wheat field | 2017.05 | Fenoxaprop-P-ethyl $\geq$ 15 Mesosulfuron-methyl $\geq$ 7 |
| HN-06 | S [b] | Hexiao | Zhumadian | Henan | N32.707560 E114.299764 | Uncultivated land | 2017.05 | Never applied |

[a] R, resistant. [b] S, susceptible.

### 2.2. Whole-Plant Dose–Response Assay and the Effect of P450s Inhibitor Malathion on Mesosulfuron-Methyl Resistance

Seed germination was carried out as described by Ge et al. [24]. Placing the pot in the artificial climate room (Model RXZ, Ningbojiangnan Instrument Factory, Ningbo, China), and keeping at 25/20 °C day/night temperatures with a 12-h cycle, 434.3 μmol·m$^{-2}$·s$^{-1}$ light intensity, and 75% relative humidity. After the seeds emerge, 10 plants of similar size and growth are reserved in each pot, and watering and fertilization shall be consistent with those before. When the plant reached the three to four leaf stage, the HN-14 population was applied with mesosulfuron-methyl at doses of 3, 9, 27, 81, 243 and 729 g a.i. ha$^{-1}$, using a compressed air moving nozzle cabinet sprayer equipped with one Teejet 9503 EVS (Greenman Machinery Company, Beijing, China) flat fan nozzle delivering 450 L·ha$^{-1}$ at 275 kPa. And the HN-06 population was applied with mesosulfuron-methyl at doses of 0.1, 0.3, 1, 3, 9, and 27 g a.i. ha$^{-1}$. After 21 days, all shoots above the ground of weeds were collected and the dry weight of each treatment was measured.

In addition, according to the method described by Liu et al. [25], in order to determine how the sensitivity of S and R populations to mesosulfuron-methyl would change when malathion is added as pretreatment, the whole-plant response experiment was conducted. When malathion with dosage of 1000 g a.i. ha$^{-1}$ is applied alone, the growth of weed seedlings is normal

### 2.3. DNA Extraction and PCR Amplification

To prove the plants from HN-14 were resistant, seedlings at the 3- to 4-leaf stage were dealt with mesosulfuron-methyl at 9 g a.i. ha$^{-1}$. 100 mg of above ground tender tissue was collected from each surviving individual from each of the HN-14 and untreated HN-06 populations at 21 days later and stored at −80 °C. DNA was extracted from plants by cetyltrimethylammonium bromide (CTAB) method [26].

The gene fragment of black-grass was amplified by Polymerase chain reactions (PCRs) is ~1800 bp. And the fragment containing all known ALS resistance mutation sites. Pairs of primers (forward primer [ALS-F]: 5′-CGTCGCCTTACCCAAACCTAC-3′; reverse primer [ALS-R]: 5′- RTCCTGCCATCACCWTCCA-3′) were designed using Primer Premier 5.0 software (Primer Premier 5.0, Biosoft, Palo Alto, CA, USA) based on the ALS gene from black-grass (AJ437300.2). The forward and reverse sequencing of purified products was completed by a commercial sequencing company (Sangon Biotech, Shanghai, China), and the sequence data obtained from sequencing were analyzed and compared using DNAMAN software(DNAMAN version 5.2.2, Lynnon Biosoft, Vandreuil, QC, Canada).

### 2.4. ALS Enzyme Activity Assay

According to Yu et al. [27], an in vitro ALS activity assay was performed, with minor modifications. The preserved fresh plant tissues were used for the extraction of ALS protein. The protein concentration in the enzyme extracts was measured using the Bradford method [28]. ALS activity was expressed as the number of nmol of acetylacetone formed per minute per milligram of protein used for the determination. The concentration of acetoin was measured by spectrophotometry (Epoch™, BioTek Instruments, Inc., Winooski, VT, USA) at 530 nm, and the results were adjusted by the standard curve generated by commercial acetoin (99%, Aladdin, Shanghai, China).

*2.5. Sensitivity to Other Herbicides with Various Modes of Action*

In this study, the susceptibility of R to a variety of herbicides with different action modes was fathomed. When seedlings at the 3- to 4-leaf stage, the herbicide dosages were applied according to Table 2. After 21 d, the above ground materials were weighed and recorded. There were 5 plants in each herbicide treatment and 3 replicates in each population. The entire experiment was conducted twice. According to the method of Moss et al. [29], the sensitivity of the R and S populations to different types of herbicides was classified. Weed samples were divided into four resistance categories (RRR, RR, R?, and S) according to the percentage of dry weight reduction.

**Table 2.** The herbicides used in cross-resistance identification and their application rates.

| Group | Herbicide | Rate (g a.i. ha$^{-1}$) | Resistance Classification [a] |
|-------|-----------|-------------------------|-------------------------------|
| APP | Fenoxaprop-P-ethyl | 62.1 | RRR |
| | Clodinafop-propargyl | 45 | RRR |
| | Diclofop-methyl | 972 | RR |
| | Fluazifop-P-butyl | 135 | RRR |
| | Haloxyfop-methyl | 40.5 | RRR |
| | Quizalofop-P-ethyl | 52.5 | S |
| CHD | Sethoxydim | 150 | RR |
| | Tralkoxydim | 390 | S |
| DEN | Pinoxaden | 45 | R? |
| ALS | Mesosulfuron-methyl | 9 | RRR |
| | Imazethapyr | 100 | S |
| PSII | Isoproturon | 900 | S |

[a] Classification of weed sensitivity to different herbicides. RRR and RR: resistance, reduce sensitivity to herbicides; R?: weed resistance is probably developing; S: susceptible.

*2.6. Statistical Analyses*

The repeated processing group data were analyzed by ANOVA (SPSS v.20.0; IBM Corporation, Armonk, NY, USA). The results show that the difference was not significant. According to the following equation, the summarized data were used the nonlinear curve for analysis by SigmaPlot software (SigmaPlot v.14.0; Systat Software, Inc., San Jose, CA, USA).

$$Y = C + \frac{D - C}{1 + \left(\frac{x}{\mathrm{ED}_{50}}\right)^b}$$

where $C$ is the lower limit, $D$ is the upper limit, and $\mathrm{ED}_{50}$ is the herbicide dose or concentration resulting in 50% growth inhibition ($GR_{50}$) or 50% reduction of ALS activity ($I_{50}$). The fitted equation was used to estimate the $GR_{50}$ value. The resistance index (RI) value was calculated by the ratio of $GR_{50}$ value of resistant population (HN-14) and susceptible population (HN-06).

**3. Results**

*3.1. Sensitivity Bioassay to Mesosulfuron-Methyl*

The sensitivity of the HN-06 population was confirmed again by the whole-plant dose–response experiments (Figure 1, Table 3). Compared with HN-06 of S population, the HN-14 population was less sensitive to mesosulfuron-methyl The $GR_{50}$ values of mesosulfuron-methyl for the HN-14 and HN-06 populations were 45.59 ($\pm$9.92) and 1.38 ($\pm$0.3) g a.i. ha$^{-1}$, respectively. According to the RI, the resistance of the blackgrass HN-14 population to mesosulfuron-methyl was 33-fold higher before malathion was applied. When mesosulfuron-methyl was applied together with malathion, the $GR_{50}$ value of mesosulfuron-methyl in the HN-14 population changed by 9.3%, from

45.59 to 50.25 g a.i. ha$^{-1}$. In contrast, the GR$_{50}$ value of the HN-06 population changed by 10.1%, from 1.38 g a.i. ha$^{-1}$ to 1.24 g a.i. ha$^{-1}$ (Figure 1, Table 3). Inhibitor (malathion) pre-treatment on GR$_{50}$ difference is not obvious. In this study, we observed that the resistance of HN-14 population to mesosulfuron-methyl is unlikely to be caused by P450.

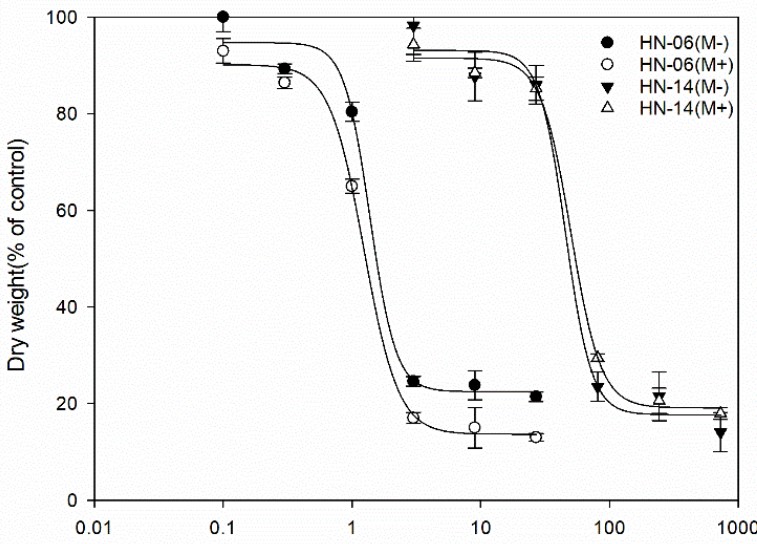

**Figure 1.** Dose–response curves for dry weight (percent of control) of the resistant (HN-14) and susceptible (HN-06) populations treated with a range of mesosulfuron-methyl doses. M-, absence of malathion pretreatment; M+, presence of malathion at 1000 g ai ha$^{-1}$. Each data point represents the mean percentage survival $\pm$ SEM of three replicate treatments.

**Table 3.** GR$_{50}$ values of S and R black-grass populations in response to mesosulfuron-methy.

| Herbicide | Population | C [a] | D [b] | B [c] | GR$_{50}$ [d] | RI [e] |
|---|---|---|---|---|---|---|
| Mesosulfuron-methyl | HN-06 | 22.4664 (3.88) | 94.7370 (3.88) | −4.3322 (2.62) | 1.38 (0.3) | |
| | HN-14 | 17.6951 (4.57) | 93.1030 (4.57) | −4.2756 (1.63) | 45.59 (9.92) | 33.0 |
| Malathion + Mesosulfuron-methyl | HN-06 | 13.6476 (2.28) | 90.1691 (2.34) | −3.2643 (0.92) | 1.24 (0.10) | |
| | HN-14 | 19.1257 (2.24) | 91.5331 (2.23) | −3.7303 (0.67) | 50.25 (5.01) | 40.52 |

[a] C, lower limit; [b] D, the upper limit; [c] b, slope of curve in GR$_{50}$; [d] GR$_{50,}$ the herbicide dose or concentration resulting in 50% growth inhibition; [e] RI resistance index (RI) value was calculated by the ratio of GR50 value of resistant popula-tion (HN-14) and susceptible population (HN-06).

### 3.2. ALS Gene Sequencing and Sequences Analysis

The 1841-bp ALS gene fragment which included the five known conserved domains was amplified. We compared the sequences with known black-grass sequences using BLAST (GenBank accession AJ437300.2)(Biological Local Alignment Search Tool). The results showed that the sequence from HN-14 population had 98.28% sequence homology with the ALS gene from black-grass (AJ437300.2). The gene sequences of the ALS fragments were aligned with the documented ALS sequence of *Arabidopsis thaliana* (NM-114714.2). In the HN-14 population, 16 surviving plants were randomly selected and sequenced. As Figure 2 shows, plants of the HN-14 population were found to hold Trp-574-Leu homozygous mutations in the ALS, which was caused by TTG to TGG nucleotide changes. In addition, among all the test plants in the HN-06 and HN-14 populations, other known amino acid mutations were not found in five known conservative domains.

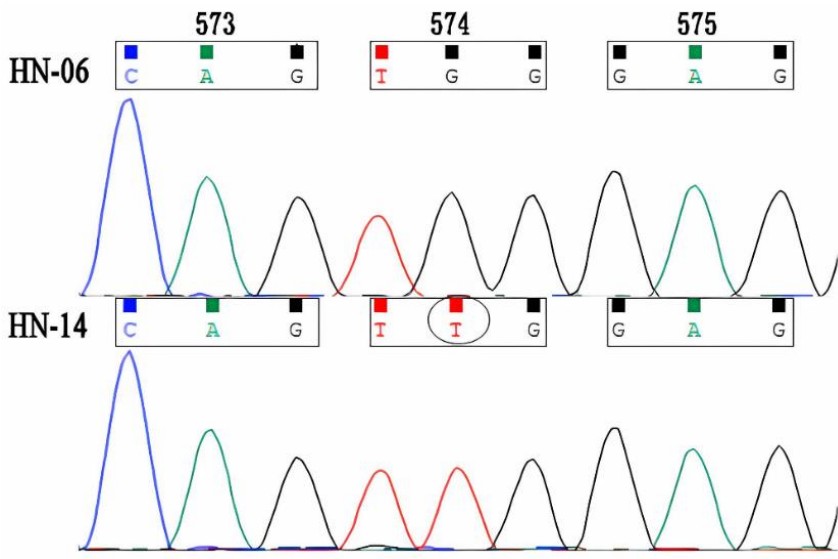

**Figure 2.** Spectrogram of sequence of ALS from resistant (HN-14) and susceptible (HN-06) black-grass. HN-14 is Trp-574-Leu homozygous mutation, and HN-06 is no mutation.

### 3.3. In Vitro ALS Inhibition Assay

In order to verify whether the sensitivity of ALS enzyme between the HN-12 and HN-06 black-grass populations is different, in vitro ALS activity was analyzed with adding or not adding mesosulfuron-methyl. The difference between the extractable ALS activity in the HN-14 population of black-grass and that of the susceptible population HN-06 was not significant when mesosulfuron-methyl was not added. The specific activities were $5.46 \pm 0.04$ and $5.74 \pm 0.02$ nmol acetoin $mg^{-1}$ protein $min^{-1}$ for HN-06 and HN-14, respectively (Figure 3a). The ALS activity of S population was significantly inhibited upon addition of messulfuron-methyl, and its $I_{50}$ value was 0.0111 μM (Figure 3b). In contrast, mutant ALS showed a greatly reduced sensitivity to mesosulfuron-methyl with the $I_{50}$ value of 0.2577 μM, which was 23.22-fold higher than that in S population.

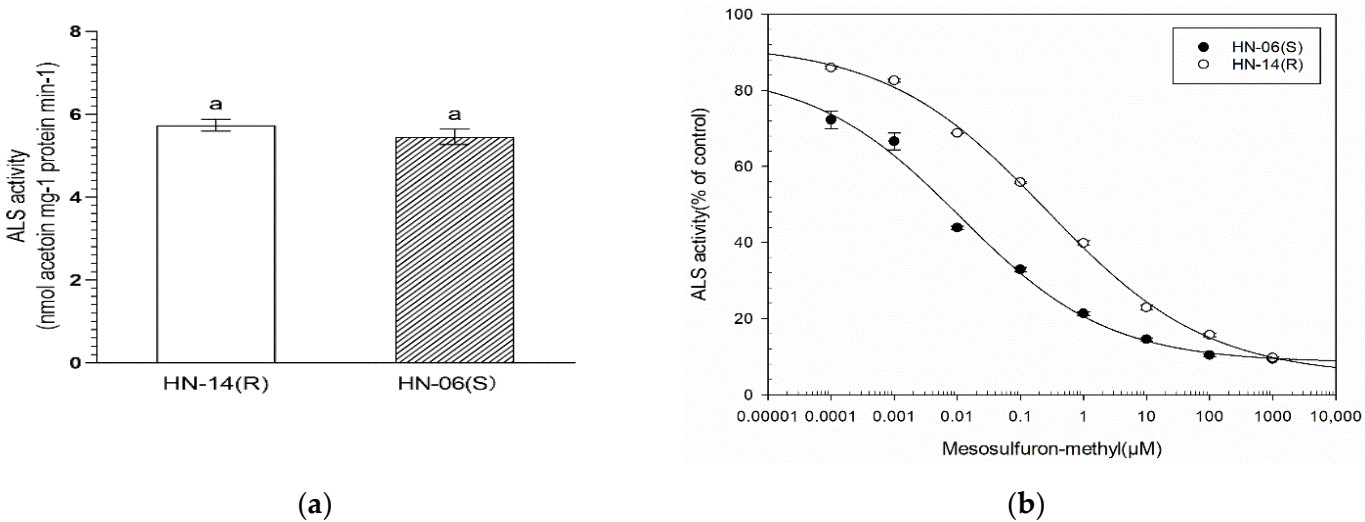

(**a**)

(**b**)

**Figure 3.** Characteristics of ALS measured from partially purified enzyme extracts from three- to four-leaf-stage ALS-herbicide-susceptible (ALS wild type), and ALS-resistant (574-Leu) black-grass plants. (**a**) Extractable total activity of R and S plants in vitro ALS. (**b**) Inhibition assay of ALS isolated from the R and S black-grass by mesosulfuron-methyl. Different letters indicate significant differences in mean estimates according to Dunnett's test ($\alpha = 0.05$).

### 3.4. Cross Resistance to Other Herbicides

To determine the sensitivity to other herbicides, one single-dose resistance screening was conducted in this study. Nine ACCase inhibitors, two ALS inhibitors, and isoproturon were used at their recommended field rates (shown in Table 2). The susceptible population HN-06 was susceptible to all applied herbicides at the recommended field dose. According to the R resistance rating system, HN-14 had developed RRR and RR resistance to fenoxaprop-P-ethyl, clodinafop-propargyl, haloxyfop-methyl, and fluazifop-P-butyl, and R″ (resistance may be developing) to pinoxaden. However, it remained sensitive to imazethapyr, quizalofop-P-ethyl, tralkoxydim, and isoproturon. These results showed that the black-grass population HN-14 had developed multiple resistance to ACCase and ALS inhibitors (Figure 4, Table 2). This resistant population also risks evolving resistance to many other herbicides. Careful attention should be paid to delaying the resistance evolution and sustainable management of this troublesome weed in wheat fields.

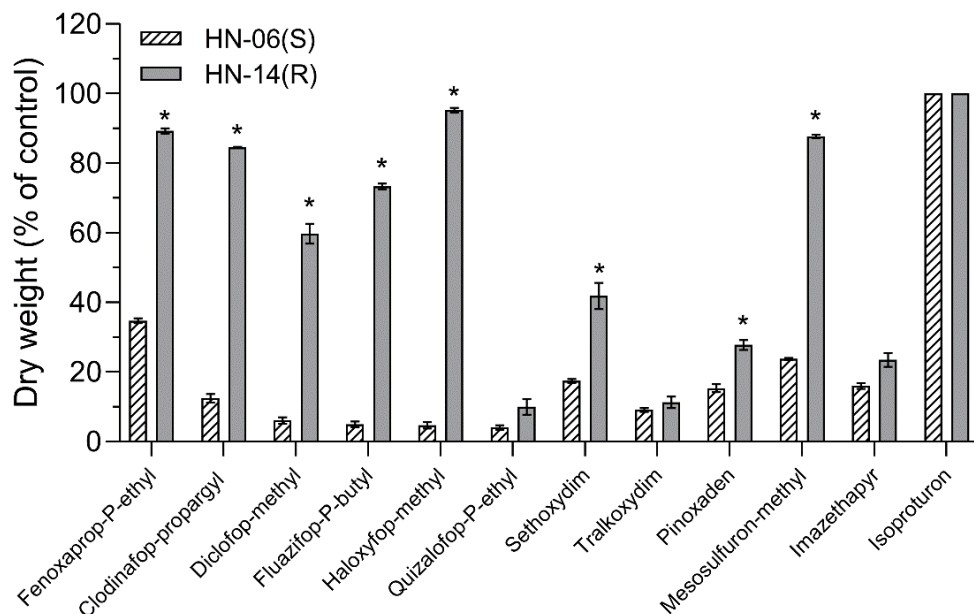

**Figure 4.** Herbicide sensitivity tests of S and R following treatments at the recommended rate. Note: *, significant difference ($p < 0.05$) between the R and S populations. Bars are means $\pm$ SEM.

### 4. Discussion

Many studies have shown that the choice to use ALS inhibitors in continuous application for three to five years can quickly generate weed resistance [12,30]. The HN-14 population was collected in a wheat field where chemical herbicides were widely used continuously for more than 7 years to control weeds. Therefore, the resistance evolution of black-grass to mesosulfuron-methyl was attributed to the selection pressure produced by herbicide stress, which resulted from the historical application of ALS-inhibiting herbicides.

In weed species collected from arable land, a variety of gene mutations have been identified, which make TSR the cause of resistance of R population to mesosulfuron-methyl [12,31]. TSR is one of the most common resistance mechanisms, which endows weed plants with high levels of resistance to different herbicides, whereas ALS resistance mutations differ between resistant weed species [31]. As reported by other weeds with resistance to ALS Inhibitors, the Trp-574-Leu mutation in the HN-14 population was confirmed resistant to mesosulfuron-methy [32]. Therefore, the Trp-574-Leu mutation in ALS is an important reason for the resistance of the black-grass HN-14 population to mesosulfuron-methy.

To demonstrate mesosulfuron-methyl resistance associated with the Trp-574-Leu substitution in the target ALS observed in the R population, ALS enzymes from the HN-14 and HN-06 populations were extracted and analyzed. The difference between the

extractable ALS activity in the HN-14 population of black-grass and that of the susceptible population HN-06 was not significant. At present, many studies have shown that there is no apparent relativity between the resistance phenotype and the total extractable ALS activity of plants, and that ALS activity may increase, decrease, or have no difference [33]. When adding mesosulfuron-methyl, the ALS activity of HN-06 population was significantly inhibited, and its $I_{50}$ value was 0.0111 μM. On the contrary, the sensitivity of mutant ALS to mesosulfuron-methyl was greatly reduced, with an $I_{50}$ value 23.22-fold higher than that in S population, which was identical with the results of bioassay of whole-plant herbicides. This was further confirms that Trp-574-Leu mutation is an important reason for the resistance of the black-grass HN-14 population to mesosulfuron-methy, which is similar to the findings of other studies of ALS-inhibitors-resistant weed species [34–37].

NTSR is another important part of the herbicide resistance mechanism [31,38]. Many studies have also reported that resistance of black-grass to ACCase and ALS inhibitors is driven by the enhancement of herbicide metabolism, with metabolic enzymes P450s, GSTs, GTS, and ABC transporters being the most common detoxification enzymes [13,14,22,31,39]. In this current study, before and after pretreatment with P450s activity inhibitor malathion, there was no difference in the sensitivity of HN-14 population to mesosulfuron-methyl. This phenomenon suggested that P450s might not participate in the resistance to mesosulfuron-methyl. However, possible involvement of GST-mediated resistance needs further work to elucidate.

In this study, the cross-resistance pattern of the specific resistant population HN-14 was also characterized using the "R" resistance rating system as its terms of weed management. The results revealed that the R population with Trp574Leu mutations exhibited resistance to mesosulfuron-methyl (SU), but was still sensitive to imazethapyr (IMI). It is somewhat different in this resistance pattern and the most other known reports. It is known that the cross-resistance patterns were various with the ALS resistance mutations. Normally, the mutations of Pro197 make weeds resistant to SUs and TPs; the mutations of Ala122, Ser653 and Gly654 promote IMIs resistance more obviously; and the mutations of Asp376 and Trp574 have broad-spectrum resistance to all five types of ALS herbicides (SUs, TPs, IMIs, PTBs, and SCTs) [32,40]. Yu and Powles [31] reported that the cross-resistance generated by the mutation of the target site does not depend on the base substitution position, the specific substitution, but also depends on the type of inhibitor, and sometimes depends on the type of weed. Yu et al. [41] also reported that produced by certain ALS mutation overview of ALS herbicide cross resistance model for cannot be based on the response of specific ALS herbicide chemistry to one or two herbicides. Yu et al. [41] reported that the homozygous plants of *Raphanus raphanistrum* with Asp-376-Glu mutation had a resistance level of more than 100-fold higher to SUs and TPs herbicides, and moderate resistance (<10-fold) to IMI herbicides imazethapyr and imazamox, but they were extremely sensitive to the IMI herbicide imazapyr. When Asn mutation occurs in yeast ALS Asp-376 and ALa mutation occurs in tobacco ALS Asp-376, ALS is difficult to be sensitive to SUs herbicide, but more sensitive to IMIs herbicide [42,43]. This phenomenon shows that when Asp-376 mutates, there will be negative interaction with IMIs herbicide. Therefore, we think that the sensitivity of HN-14 to imazethapyr may be explained by the diversity of weed species, and that the mutant form of Trp-574 in black-grass may also negatively interact with IMI herbicides or with imazethapyr. The evolution of specific herbicide resistance necessitates further research.

Here, the sensitivity of HN-14 to other herbicides (ACCase and PSII inhibitors) was also determined. The results showed that HN-14 has evolved multiple resistance to ACCase inhibitors fenoxaprop-P-ethyl (APP), haloxyfop-methyl (APP), diclofop-methyl (APP), fluazifop-P-butyl (APP), clodinafop-propargyl (APP), and sethoxydim (CHD). However, it is still sensitive to the ACCase inhibitor quizalofop-P-ethyl (APP), tralkoxydim (CHD), and has the risk of developing resistance to pinoxaden (DEN). Additionally, the HN-14 plants were very sensitive to the PSII inhibitor isoproturon. Multiple resistance of black-grass to ALS and ACCase inhibitors may be also triggered by the wide application of ALS and

ACCase inhibitors in wheat fields in China. Guo et al. [6] and Zhao et al. [44] reported that shortawn foxtail has evolved high resistance to mesosulfuron-methyl. Bi et al. [20] reported that *A. japonicus* has evolved multiple resistance to ACCase- and ALS-inhibiting herbicides. Lan et al. [22] reported that a black-grass population was multiple-resistant to pinoxaden and pyroxsulam, and cross-resistant to mesosulfuron-methyl. Cross-resistance dominated by target gene mutation is not only related to the mutation form, but also to weed species, homozygosity and/or heterozygosity of mutant plants, number of alleles, herbicide, and the type of herbicide and the dosage applied have a certain relationship [45–47]. For example, the Asp-2078-Gly mutation is easy to produce high-level resistance to ACCase herbicide, but sensitive to CHD herbicide clethodim [46,47]. Ile-1781-Thr mutant black-grass plants were moderately resistant to clodinafop-propargyl and cycloxydim, but were very sensitive to pinoxaden and clethodim [15]. Evidence shows that ACCase (APP group) herbicides are bound in a domain close to the catalytic site and partially overlapping the catalytic site [48]. Nevertheless, the exact binding details of the some ACCase herbicides across three chemical groups (APP, CHD and PPZ) are still unknown. In this study, the HN-14 population presented different cross-resistance to six ACCase inhibitors, which all belonged to the APP chemical group. The HN-14 plants also exhibited resistance to sethoxydim (CHD), which is relatively less identified in ACCase inhibitors resistance cases. It is wrong to assume that all weeds have a high level of resistance to herbicides due to mutations in target sites. The substitution of some amino acids may prevent the combination with herbicides, but will not affect the binding of acetyl-CoA, the substrate at the catalytic site, while other mutations overlapping with the catalytic site may adversely affect the binding of acetyl-CoA, thus affecting the function of ACCase [49]. It is not clear whether the black-grass HN-14 population has such an adverse effect that makes it sensitive to quizalofop-P-ethyl. This phenomenon may be caused by specific TSR mutations and/or NTSR mechanism, and this requires further investigation to clarify.

As the TSR and NTSR have also been identified in other resistant black-grass populations, the multiple herbicide resistance in this weed should be given attention, and diverse control methods are encouraged to suppress further distribution and development. Crop rotation is a very effective nonchemical strategy for weed management. When the density of black-grass is low, spring planting of short-term profit losses will be far greater than the reduction of the profit of the use of herbicides and increasing production-related profits. However, as a long-term strategy, crop rotation will reduce the density of black-grass for a prolonged time [50]. Lutman et al. [9] reported that when the planting of winter wheat was postponed from September to the end of October, the density of black-grass was reduced by about 50%, and selecting more competitive varieties may reduce black-grass by 22%. When herbicides are combined with nonchemical practices, weed management is the most effective [11]. This strategy should be actively adopted and used to delay the generation of black-grass resistance, and to extend the sustainability of PSII inhibitor isoprotection, which is still very effectively control the black-grass in the amount of arable land.

## 5. Conclusions

This study reported that one black-grass population has evolved resistance to ALS inhibitor mesosulfuron-methyl, which was endowed by Trp-574-Leu resistance mutation in target ALS. The results also indicated that this population has evolved multiple resistance to six ACCase inhibitors (five APPs and one CHD). However, this resistant population was still sensitive to imazethapyr (IMI), quizalofop-P-ethyl (APP), tralkoxydim (CHD), and the PSII inhibitor isoproturon. Therefore, in order to delay the resistance of black grass, the combination of isoproturon can provide a more intelligent strategy. For this reason, it is still encouraged to adopt various methods for integrated weed management.

**Author Contributions:** Conceptualization, W.L. and J.W.; methodology, X.Q. and H.W.; validation, X.Q., C.Y. and M.H.; formal analysis, X.Q.; investigation, X.Q., C.Y. and Y.D.; resources, W.L., N.Z. and J.W.; data curation, X.Q. and Y.D.; writing—original draft preparation, X.Q.; writing—review and editing, W.L. and H.W.; visualization, X.Q.; supervision, W.L., J.W. and H.W.; project administration, W.L. All authors have read and agreed to the published version of the manuscript.

**Funding:** This research was funded by the Key R&D Program of Shandong Province (2021CXGC010811), and the Project ZR2021MC030 supported by Shandong Provincial Natural Science Foundation.

**Conflicts of Interest:** The authors declare no conflict of interest.

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
