# Peer review of "Molecular Basis of Resistance to Mesosulfuron-Methyl in a Black-Grass (Alopecurus myosuroides Huds.) Population from China"

_agronomy, doi:10.3390/agronomy12092203_

Round 1

Reviewer 1 Report

In this study, the authors examined the resistance of a black-grass (Alopecurus myosuroides) population to mesosulfuron-methyl and other herbicides. Although several papers have been published on this topic, further research and recording of weed populations with resistance to various herbicides is important. This article is well written and presented useful experimental results that deserved to published after minor revision.

Comments

Abstract: This part is well written.

Introduction: This section is well written and sufficient information is presented that highlights the importance of the study.

Line 64: the scientific name of Alopecurus should be written in Italics.

Material and Methods: This section needs minor revision.

Line 87: The date of seeds collection should be added.

Lines 95-96: The date of sowing should be also added.

Line 95: The part “Ge et al. [20]. And germination seeds” should be revised e.g. “Ge et al. [20] and germinated seeds”.

Line 112: “leafstage” should be corrected to “leaf stage”.

Line 113: “21DAT” should be corrected to “21 DAT”.

Line 114. The space prior to words “were conducted” should be shorter.

Figure 2: The quality of this figure should be improved.

Results: This section is well written.

Discussion: This parts is needs minor revision. The authors found that the black grass population HN-14 was resistant to fluazifop-p-butyl but was sensitive to quizalofop-p-ethyl. The authors should give an explanation about this result and they should support this result by adding a relevant reference if it is available.

Appendix: The Figures A1, A2, A3 and A4 are the same with figures 1 to 4.  

Line 258: “ weed specie” should be corrected to “weed species”

Line 287: Yu et al. should be corrected. The number of reference should be added and should be deleted from the end of the phrase.

References: Scientific names of plant species should be written in italics. The second word of plant species should be start with small letter.

Author Response

Thank you for giving us opportunity to revise the manuscript (agronomy-1906855). We appreciate reviewers’ comments which are helpful for us to improve the manuscript. According to comments, we have modified the manuscript carefully. Please look for the detailed changes in attached file.

Once again, thanks for your effort and time on improving our manuscript.

Reviewer 2 Report

I find this this a good article, clear and well written. Please find a few suggestions below.

I would suggest to the authors to check the keywords, they shouldn't repeat the words of the title.

Please add in the introduction a small state of the art of non-chemical weed control in wheat. It is a possible solution for resistance to herbicides.

Please add standard errors in table 3 (between brackets next to the estimated value of each parameter). Moreover may you please specify if you used mean values or replications to fit the dose response curves?

Please improve the quality of figure 2.

Please add in the discussion a small paragraph on non-chemical weed control (the same suggestion given for the introduction).

Maybe I am mistaken, but does the appendix show the same figures of the manuscript?

Thank you and gook much for the publication of your manuscript.

Author Response

Thank you for giving us opportunity to revise the manuscript (agronomy-1906855). We appreciate reviewers’ comments which are helpful for us to improve the manuscript. According to comments, we have modified the manuscript carefully. Please look for the detailed changes in the attached file.

Once again, thanks for your effort and time on improving our manuscript.
